# Generalizing and Tensorizing Subgraph Search in the Supernet

## Abstract

Recently, a special kind of graph, i.e., supernet, which allows two nodes connected by multi-choice edges, has exhibited its power in neural architecture search (NAS) by searching better architectures for computer vision (CV) and natural language processing (NLP) tasks. In this paper, we discover that the design of such discrete architectures also appears in many other important learning tasks, e.g., logical chain inference in knowledge graphs (KGs) and meta-path discovery in heterogeneous information networks (HINs). Thus, we are motivated to *generalize the supernet* search problem on a broader horizon. However, none of the existing works are effective since the supernet's topology is highly task-dependent and diverse. To address this issue, we propose to *tensorize the supernet*, i.e., unify the subgraph search problems by a tensor formulation and encode the topology inside the supernet by a tensor network. We further propose an efficient algorithm that admits both stochastic and deterministic objectives to solve the search problem. Finally, we perform extensive experiments on diverse learning tasks, i.e., architecture design for CV, logic inference for KG, and meta-path discovery for HIN. Empirical results demonstrate that our method leads to better performance and architectures.

## 1 Introduction

Deep learning (Goodfellow et al., 2017) has been successfully applied in many applications, such as image classification for computer vision (CV) (LeCun et al., 1998; Krizhevsky et al., 2012; He et al., 2016; Huang et al., 2017) and language modeling for natural language processing (NLP) (Mikolov et al., 2013; Devlin et al., 2018). While the architecture design is of great importance to deep learning, manually designing proper architectures for a certain task is hard and requires lots of human efforts or sometimes even impossible (Zoph & Le, 2017; Baker et al., 2016).

Recently, neural architecture search (NAS) techniques (Elsken et al., 2019) have been developed to alleviate this issue, which mainly focuses on CV and NLP tasks. Behind existing NAS methods, a multi-graph (Skiena., 1992) structure, i.e., supernet (Zoph et al., 2017; Pham et al., 2018; Liu et al., 2018), where nodes are connected by edges with multiple choices, has played a central role. In such context, the choices on each edge are different operations, and the subgraphs correspond to different neural architectures. The objective here is to find a suitable subgraph in this supernet, i.e. better neural architectures for a given task.

However, the supernet does not only arise in CV/NLP field and we find it also emerge in many other deep learning areas (see Table 1). An example is logical chain inference on knowledge graphs (Yang et al., 2017; Sadeghian et al., 2019; Qu & Tang, 2019), where the construction logical rules can be modeled by a supernet. Another example is meta-path discovery in heterogeneous information networks (Yun et al., 2019; Wan et al., 2020), where the discovery of meta-paths can also be modeled by a supernet. Therefore, we propose to broaden the horizon of NAS, i.e., generalize it to many deep learning fields and solve the new NAS problem under a unified framework.

Since subgraphs are discrete objects (choices on each edge are discrete), it has been a common approach (Liu et al., 2018; Sadeghian et al., 2019; Yun et al., 2019) to transform it into a continuous optimization problem. Previous methods often introduce continuous parameters separately for each edge. However, this formulation cannot generalize to different supernets as the topological structures of supernets are highly task-dependent and diverse. Therefore, it will fail to capture the supernet's topology and hence be ineffective.

Table 1: A comparison of existing NAS/non-NAS works for designing discrete architectures based on our tensorized formulation for the supernet. "Topology" indicate topological structure of the supernet is utilized or not.

| | representative method | domain | supernet encoding | topology | optimization algorithm |
|---|---|---|---|---|---|
| existing NAS works | NASNet (Zoph et al., 2017) | CV | RNN | $\times$ | policy gradient |
| | ENAS (Pham et al., 2018) | CV/NLP | RNN | $\times$ | policy gradient |
| | DARTS (Liu et al., 2018) | CV/NLP | rank-1 CP | $\times$ | gradient descent |
| | SNAS (Xie et al., 2018) | CV | rank-1 CP | $\times$ | gradient descent |
| | NASP (Yao et al., 2020) | CV/NLP | rank-1 CP | $\times$ | proximal algorithm |
| non-NAS works | DRUM (Sadeghian et al., 2019) | KG | rank-$n$ CP | $\times$ | gradient descent |
| | GTN (Yun et al., 2019) | HIN | rank-1 CP | $\times$ | gradient descent |
| proposed | TRACE | all | TN | $\checkmark$ | gradient descent |

In this paper, we propose a novel method *TRACE* to introduce a continuous parameter for each *subgraph* (all these parameters will form a tensor). Then, we propose to construct a tensor network (TN) (andrzej. et al., 2016; 2017) based on the topological structure of supernet. For different tensor networks, we introduce an efficient algorithm for optimization on supernets. Extensive experiments are conducted on diverse deep learning tasks. Empirical results demonstrate that TRACE performs better than the state-of-the-art methods in each domain. As a summary, our contributions are as follows:

- We broaden the horizon of existing supernet-based NAS methods. Specifically, we *generalize* the concept of subgraph search in supernet from NAS to other deep learning tasks that have graph-like structures and propose to solve them in a unified framework by *tensorizing* the supernet.
- While existing supernet-based NAS methods ignore the topological structure of the supernet, we encode the supernet in a topology-aware manner based on the tensor network and propose an efficient algorithm to solve the search problem.
- We conduct extensive experiments on various learning tasks, i.e., architecture design for CV, logical inference for KG, and meta-path discovery for HIN. Empirical results demonstrate that our method can find better architectures, which lead to state-of-the-art performance on various applications.

## 2 RELATED WORKS

### 2.1 SUPERNET IN NEURAL ARCHITECTURE SEARCH (NAS)

There have been numerous algorithms proposed to solve the NAS problem. The first NAS work, NASRL (Zoph & Le, 2017), models the NAS problem as a multiple decision making problem and proposes to use reinforcement learning (RL) (Sutton & Barto, 2018) to solve this problem. However, this formulation does not consider the repetitively stacked nature of neural architectures and is very inefficient as it has to train many different networks to converge. To alleviate this issue, NASNet (Zoph et al., 2017) first models NAS as an optimization problem on supernet. The supernet formulation enables searching for transferrable architectures across different datasets and improves the searching efficiency. Later, based on the supernet formulation, ENAS (Pham et al., 2018) proposes weight-sharing techniques, which shares the weight of each subgraph in a supernet. This technique further improves searching efficiency and many different methods have been proposed under this framework (see Table 1), including DARTS (Liu et al., 2018), SNAS (Xie et al., 2018), and NASP (Yao et al., 2020). DARTS is the first to introduce deterministic formulation to NAS field, and SNAS uses a similar parametrized method with DARTS under stochastic formulation. NASP improves upon DARTS by using proximal operator (Parikh & Boyd, 2014) and activates only one subgraphs in each iteration to avoid co-adaptation between subgraphs.

### 2.2 TENSOR METHODS IN MACHINE LEARNING

A tensor (Kolda & Bader, 2009) is a multi-dimensional array as an extension to a vector or matrix. Tensor methods have found wide applications in machine learning, including network

compression (Novikov et al., 2015; Wang et al., 2018) and knowledge graph completion (Liu et al., 2020). Recently, andrzej. et al. (2016; 2017) have proposed a unified framework called *tensor network (TN)*, which uses an undirected graph to represent tensor decomposition methods. By using different graphs, TN can cover many tensor decomposition methods as a special case, e.g., CP (Bro, 1997), Tucker (Tucker, 1966), tensor train (Oseledets, 2011), and tensor ring decomposition (Zhao et al., 2016). However, it is not an easy task to construct a tensor network for a given problem, as the topological structure of tensor network is hard to design (Li & Sun, 2020).

## 3 PROPOSED METHOD

Here, we describe our method for the supernet search problem. Section 3.1-3.2 introduce how we *tensorize* the supernets. Section 3.3 propose an optimization algorithm for the search problem which can be utilized for both stochastic and deterministic objectives. Finally, how supernet appears beyond existing NAS works and can be *generalized* to these new tasks are presented in Section 3.4.

**Notations.** In this paper, we will use $\mathcal{S}$ to denote a supernet, and $\mathcal{P}$ to denote a subgraph in supernet. For a supernet $\mathcal{S}$ with $T$ edges, we let all edges be indexed by $e_1, \ldots, e_T$, and define $C_t$ to be the number of choices on edge $e_t$ with $t \in \{1, \ldots, T\}$. With subscript $i_1, \ldots, i_T$ denoted by $i_-$ for short, we use $\mathcal{S}_{i_-}$ to denote the subgraph with choices $i_1 \in \{1, \ldots, C_1\}, \ldots, i_T \in \{1, \ldots, C_T\}$. And we use $\mathrm{softmax}(\boldsymbol{o}_i) = \exp(\boldsymbol{o}_i)/\sum_{j=1}^n \exp(\boldsymbol{o}_j)$ to denote the softmax operation over a vector $\boldsymbol{o}_i \in \mathbb{R}^n$.

### 3.1 A TENSOR FORMULATION FOR SUPERNET

While existing works (Liu et al., 2018; Zoph et al., 2017; Pham et al., 2018) introduce parameters separately for each edge $e_t$, since edges may correlate with each other, a more general and natural approach is to introduce a continuous parameter directly for each *subgraph* $\mathcal{P} \in \mathcal{S}$. Note that a subgraph $\mathcal{P}$ can be distinguished by its choices on each edge, we propose to encode all possible choices into a tensor $\mathcal{T} \in \mathbb{R}^{C_1 \times \cdots \times C_T}$ and take these choices as indices, i.e., $i_-$, to the tensor $\mathcal{T}$. As a consequence, the architecture of the subgraph $\mathcal{P}$ is indexed as $\mathcal{S}_{i_-}$, and $\mathcal{T}_{i_-} \in [0, 1]$ represent how "good" $\mathcal{P}$ can be.

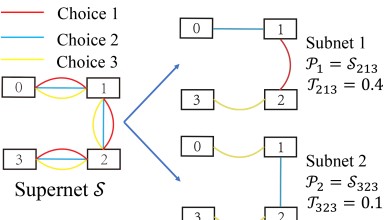

Figure 1: An example of supernet and two subgraphs with index $i_-$.

Let $f(\boldsymbol{w}, \mathcal{P})$ stand for a learning model with the parameter $\boldsymbol{w}$ and subgraph $\mathcal{P} \in \mathcal{T}$, $\mathcal{L}$ be the loss of $f$ on the training dataset $\mathcal{D}_{\mathrm{tr}}$, and $\mathcal{J}$ measure the performance of $f$ on the validation set $\mathcal{D}_{\mathrm{val}}$. Under above tensor formulation, the search objective here is:

$$\max_{\mathcal{P} \in \mathcal{T}} \mathcal{J}(f(\boldsymbol{w}^*(\mathcal{P}), \mathcal{P}); \mathcal{D}_{\mathrm{val}}), \quad \text{s.t.} \quad \begin{cases} \boldsymbol{w}^*(\mathcal{P}) = \arg\min_{\boldsymbol{w}} \mathcal{L}\left(f(\boldsymbol{w}, \mathcal{P}); \mathcal{D}_{\mathrm{tr}}\right) \\ \sum_{i_-} \mathcal{T}_{i_-} = 1 \end{cases}, \quad (1)$$

Same as existing supernet search works, the subgraph $\mathcal{P}$ is searched in the upper-level, while the network weight $\boldsymbol{w}$ is trained in the lower-level. $\mathcal{D}_{\mathrm{tr}}$ is the training set and $\mathcal{D}_{\mathrm{val}}$ is the validation set. However, we have the extra constraint $\sum_{i_-} \mathcal{T}_{i_-} = 1$ here, which is to ensure that the sum of probabilities for all subgraphs is one.

Next, we show how $\mathcal{P}$ and $\mathcal{T}$ can be parameterized with topological information from the supernet in Section 3.2. Then, a gradient-based algorithm which can effectively handle the constrained bi-level optimization problem (1) is proposed in Section 3.3.

### 3.2 ENCODING SUPERNET TOPOLOGY BY TENSOR NETWORK (TN)

Existing methods consider each edge separately and can be seen as rank-1 factorization of the full tensor $\mathcal{T}$ (see Table 1) under (1), i.e. $\mathcal{T}_{i_-} = \theta_{i_1}^1 \ldots \theta_{i_T}^T$ where $\theta_{i_t}^t \in \mathbb{R}$ is the continuous parameter for choice $i_t$ on edge $t$. However, this formulation ignores the topological structure of different supernets, as it uses the same decomposition method for all supernets. Motivated by such limitation, we propose to introduce tensor network (TN) to better encode the topological structure of supernet. Our encoding

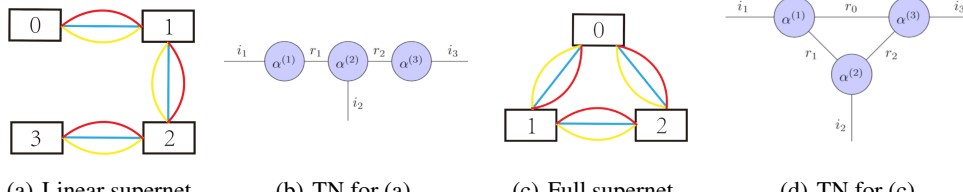

(a) Linear supernet.  (b) TN for (a).  (c) Full supernet.  (d) TN for (c).

Figure 2: Some examalphaple of supernets and the corresponding tensor networks (TNs). Information flows in supernet (a) (c) follow the order of number in blocks. Circles in (b) and (d) represent core tensors and edges represent indices. Edges connected by two circles indicate index summation on $r_{N_1(t)}$ or $r_{N_2(t)}$, where $N_1(t)$ (or $N_2(t)$) is the index of nodes in supernet (1, 2 in (a) and 0, 1, 2 in (c)). The formula of (b) is $\mathcal{T}_{i_1,i_2,i_3} = \sum_{r_1,r_2} \alpha^1_{i_1,r_1} \alpha^2_{r_1,i_2,r_2} \alpha^3_{i_3,r_2}$ and the formula of (d) is $\mathcal{T}_{i_1,i_2,i_3} = \sum_{r_0,r_1,r_2} \alpha^1_{r_0,i_1,r_1} \alpha^2_{r_1,i_2,r_2} \alpha^3_{r_2,i_3,r_0}$.

process is described in Algorithm 1, where $\mathcal{N}(\mathcal{S})$ to denotes the set of nodes in this supernet and $\mathcal{N}'(\mathcal{S}) \subseteq \mathcal{N}(\mathcal{S})$ denotes the set of nodes that are connected to more than one edge.

Specifically, we introduce a third-order tensor $\alpha^t$ for each edge, which is based on previous methods (e.g., DARTS and SNAS), but uses tensors instead of vectors to allow more flexibility. And we introduce hyper-parameters $R_{N_1(t)}$ and $R_{N_2(t)}$, which are corresponding to the rank of tensor network. Then, we use index summation to reflect the topological structure (common nodes) between different edges, i.e.,

$$\mathcal{T}_{i_-} = \sum_{r_n, n \in \mathcal{N}'(\mathcal{S})}^{R_n} \prod_{t=1}^{T} \alpha^t_{r_{N_1(t)}, i_t, r_{N_2(t)}}, \tag{2}$$

We also give two examples in [1]Figure 2 to illustrate our tensorizing process for two specific supernets.

---

**Algorithm 1** Supernet encoding process (a step-by-step and graphical illustration is in Appendix.G).

**Input:** Supernet $\mathcal{S}$;
  1: Introduce $\alpha^t \in \mathbb{R}^{R_{N_1(t)} \times C_t \times R_{N_2(t)}}$ for edge $e_t$ which connects nodes $N_1(t)$ and $N_2(t)$;
  2: Remove isolated nodes and obtain $\mathcal{N}'(\mathcal{S})$;
  3: Compute $\mathcal{T}_{i_-}$ by (2);
  4: **return** encoded supernet $\mathcal{T}_{i_-}$;

---

The reason of using $\mathcal{N}'(\mathcal{S})$ instead of $\mathcal{N}(\mathcal{S})$ is [2]Proposition 1, which shows that using $\mathcal{N}'(\mathcal{S})$ will not restrict the expressive power but allows the usage of less parameters.

**Proposition 1.** *Any tensors that can be expressed by $\mathcal{T}_{i_-} = \sum_{r_n, n \in \mathcal{N}(\mathcal{S})}^{R_n} \prod_{t=1}^{T} \alpha^t_{r_{N_1(t)}, i_t, r_{N_2(t)}}$ can also be expressed by $\mathcal{T}_{i_-} = \sum_{r_n, n \in \mathcal{N}'(\mathcal{S})}^{R_n} \prod_{t=1}^{T} \alpha^t_{r_{N_1(t)}, i_t, r_{N_2(t)}}$.*

### 3.3 Search Algorithm

To handle subgraph search problems for various applications (Table 1), we need an algorithm that can solve the generalized supernet search problem in (1), which is parameterized by (2). However, the resulting problem is hard to solve, as we need to handle the constraint on $\mathcal{T}$, i.e., $\sum_{i_-} \mathcal{T}_{i_-} = 1$. To address this issue, we propose to re-parameterize $\alpha$ in (2) using softmax trick, i.e.,

$$\bar{\mathcal{T}}_{i_-} = 1/\prod_{n \in \mathcal{N}'(\mathcal{S})} R_n \sum_{r_n, n \in \mathcal{N}'(\mathcal{S})}^{R_n} \prod_{t=1}^{T} \text{softmax}(\beta^{(t)}_{r_{N_1(t)}, i_t, r_{N_2(t)}}). \tag{3}$$

Then, the optimization objective $\mathcal{J}$ in (1) becomes

$$\max_\beta \mathcal{J}_\beta(f(\boldsymbol{w}^*(\beta), \mathcal{P}(\beta)); \mathcal{D}_{\text{val}}), \quad \text{s.t.} \quad \boldsymbol{w}^*(\beta) = \arg\min_{\boldsymbol{w}} \mathcal{L}(f(\boldsymbol{w}, \mathcal{P}(\beta)); \mathcal{D}_{\text{tr}}). \tag{4}$$

---

[1]Figure 2(b) and (d) also follow the diagram figure of tensor network, reader may refer (andrzej. et al., 2017) for more details.

[2]All proofs are in Appendix D.

where we have substitute discrete subgraphs $\mathcal{P}$ and normalization constraint on $\mathcal{T}$ with continuous parameters $\beta$. As shown in Proposition 2, we can now solve a unconstrained problem on $\beta$, while keeping the constraint on $\mathcal{T}$ satisfied.

**Proposition 2.** $\sum_{i_-} \bar{\mathcal{T}}_{i_-} = 1$, where $\bar{\mathcal{T}}$ is given in (3).

Thus, gradient-based training strategies can be reused, which makes the optimization very efficient. The complete steps for optimizing (4) are presented in Algorithm 2. Note that, our algorithm can solve both deterministic and stochastic formulation (see Appendix A). After the optimization of $\beta$ is converged, we obtain $\mathcal{P}^*$ from tensor $\bar{\mathcal{T}}$ and retrain the model to obtain $\boldsymbol{w}^*$.

---

**Algorithm 2** TRACE: Proposed subgraph search algorithm.

---

**Input:** A *subgraph search problem* with training set $\mathcal{D}_{\text{tr}}$, validation set $\mathcal{D}_{\text{val}}$ and supernet $\mathcal{S}$;
 1: Tensorize the supernet $\mathcal{T}$ with Algorithm 1;
 2: Re-parameterize $\mathcal{T}$ to $\bar{\mathcal{T}}$ using (3);
 3: **while** not converged **do**
 4:     Obtain a subgraph $\mathcal{P}$ from $\bar{\mathcal{T}}$ (deterministic or stochastic);
 5:     Solve $\boldsymbol{w}^*(\beta) = \arg\min_{\boldsymbol{w}} \mathcal{L}\left(f(\boldsymbol{w}, \beta), \mathcal{D}_{\text{tr}}\right)$;
 6:     Update supernet parameters $\beta$ by gradient ascending $\nabla_\beta \mathcal{J}_\beta$;
 7: **end while**
 8: Obtain $\mathcal{P}^* = \mathcal{S}_{i_-}$ from the final $\bar{\mathcal{T}}$ by setting $i_- = \arg\max_{i_-} \bar{\mathcal{T}}_{i_-}$;
 9: Obtain $\boldsymbol{w}^*(\mathcal{P}^*) = \arg\min_{\boldsymbol{w}}; \mathcal{L}\left(f(\boldsymbol{w}, \mathcal{P}^*), \mathcal{D}_{\text{tr}}\right)$ by retraining $f(\boldsymbol{w}, \mathcal{P}^*)$;
10: **return** $\mathcal{P}^*$ (searched architecture) and $\boldsymbol{w}^*$ (fine-tuned parameters);

---

### 3.4 Subgraph Search beyond Existing NAS

Despite NAS, there are many important problems in machine learning that have a graph-like structure. Examples include meta-path discovery (Yang et al., 2018; Yun et al., 2019; Wan et al., 2020), logical chain inference (Yang et al., 2017; Sadeghian et al., 2019) and structure learning between data points (Franceschi et al., 2019). Inspired by the recent work that exploits graph-like structures in NAS (Li et al., 2020; You et al., 2020), we propose to model them also as a subgraph search problem on supernets.

**Meta-path discovery.** Heterogeneous information networks (HINs) (Sun & Han, 2012; Shi et al., 2017) are networks whose nodes and edges have multiple types. HIN has been widely used in many real-world network mining scenarios, e.g., node classification (Wang et al., 2019) and recommendation (Zhao et al., 2017). For a heterogeneous network, a meta-path (Sun et al., 2011) is a path defined on it with multiple edge types. Intuitively, different meta-paths capture different semantic information from a heterogeneous network and it is important to find suitable meta-paths for different applications on HINs. However, designing a meta-path on a HIN is not a trivial task and requires much human effort and domain knowledge (Zhao et al., 2017; Yang et al., 2018). Thus, we propose to automatically discover informative meta-paths instead of to manually design.

To solve the meta-path discovery problem under the supernet framework, we first construct a supernet $\mathcal{S}$ (see Figure 2(a)) and a subgraph $\mathcal{P} \in \mathcal{S}$ will be a meta-path on HIN. While GTN (Yun et al., 2019) introduces weights separately for each edge, our model $f(\boldsymbol{w}, \mathcal{P})$ uses a tensor $\mathcal{T}$ to model $\mathcal{P}$ as a whole. The performance metric $\mathcal{L}(\cdot)$ and $\mathcal{M}(\cdot)$ will depend on the downstream task. In our experiments on node classification, we use cross-entropy loss for $\mathcal{L}(\cdot)$ and macro F1 score for $\mathcal{M}(\cdot)$.

**Logical chain inference.** A knowledge graph (KG) (Singhal, 2012; Wang et al., 2017) is a multi-relational graph composed of entities (nodes) and relations (different types of edges). KG has found wide applications in many different areas, including question answering (Lukovnikov et al., 2017) and recommendation (Zhang et al., 2016). An important method to understand semantics in KG is logical chain inference, which aims to find underlying logic rules in KG. Specifically, a logical chain is a path on knowledge graph with the following form $x \xrightarrow{}_{B_1} z_1 \xrightarrow{}_{B_2} \ldots \xrightarrow{}_{B_T} y$, where $x, y, z_1, \ldots$ are entities and $B_1, B_2, \ldots, B_T$ are different relations in the knowledge graph. And logical chain inference is to use a logical chain to approximate a relation in KG. Obviously, different logical chains can have critical influence on KG as incorrect logic rules will lead to wrong facts.

Table 2: Comparison with NAS methods in stand-alone setting on NAS-Bench-201.

| Method | CIFAR-10 | | CIFAR-100 | | ImageNet-16-120 | |
|---|---|---|---|---|---|---|
| | validation | test | validation | test | validation | test |
| ResNet (He et al., 2016) | 90.83 | 93.97 | 70.42 | 70.86 | 44.53 | 43.63 |
| Random | 90.93±0.36 | 93.70±0.36 | 70.60±1.37 | 70.65±1.38 | 42.92±2.00 | 42.96±2.15 |
| REINFORCE | 91.09±0.37 | 93.85±0.37 | 71.61±1.12 | 71.71±1.09 | 45.05±1.02 | 45.24±1.18 |
| BOHB | 90.82±0.53 | 93.61±0.52 | 70.74±1.29 | 70.85±1.28 | 44.26±1.36 | 44.42±1.49 |
| REA | 91.19±0.31 | 93.92±0.30 | 71.81±1.12 | 71.84±0.99 | 45.15±0.89 | 45.54±1.03 |
| TRACE | **91.33±0.19** | **94.20±0.17** | **73.26±0.22** | **73.29±0.20** | **46.19±0.16** | **46.19±0.15** |
| TRACE (Best) | 91.53 | **94.37** | **73.49** | **73.51** | 46.37 | 46.34 |
| Best in Bench-201 | 91.61 | 94.37 | 73.49 | 73.51 | 46.77 | 47.31 |

However, directly solving the inference problem will have a exponential complexity as we have to enumerate all relations (Hamilton et al., 2018). Thus, we propose to model it as a supernet search problem to reduce complexity.

Since logical chain has a chain structure, we construct a supernet as in Figure 2(a). Denote our target relation as $B_r$, the adjacency matrix of relation $B_r$ as $\mathbf{A}_{B_r}$, and the one-hot vector corresponding to entity $x$ as $\mathbf{v}_x$. Our learning model $f(\boldsymbol{w}, \mathcal{P})$ now has no model parameter $\boldsymbol{w}$, and the original bi-level problem is reduced to a single-level one with the following performance measure $\mathcal{M}(f(\boldsymbol{w}, \mathcal{P}), \mathcal{D}) = \sum_{B_r(x,y)=1 \text{ in } \mathcal{D}} \mathbf{v}_x^\top (\prod_{i=1}^{T} \mathbf{A}_{B_i}) \mathbf{v}_y$, which counts the number of pairs $(x, y)$ that has relation $B_r$ and is predicted by logical chain $x \longrightarrow_{B_1} z_1 \longrightarrow_{B_2} \ldots \longrightarrow_{B_T} y$ in the KG $\mathcal{D}$.

## 4 EXPERIMENTS

All experiments are implemented on PyTorch (Paszke et al., 2017) except for the logical chain inference, which is implemented on TensorFlow (Abadi et al., 2016) following DRUM (Sadeghian et al., 2019). We have done all experiments on a single NVIDIA RTX 2080 Ti GPU.

### 4.1 BENCHMARK PERFORMANCE COMPARISON

Here, we compare our proposed method with the state-of-the-art methods on three different applications that can be seen as subgraph search problems, i.e., neural architecture design for image classification, logical chain inference from KG, and meta-path discovery in HIN.

### 4.1.1 DESIGNING CONVOLUTIONAL NEURAL NETWORK (CNN) ARCHITECTURES

We first apply TRACE to the architecture design problem on CNN for image classification, which is currently the most famous application for supernet-based methods. We consider the following two different settings for our NAS experiments: (i). Stand-alone (Zoph & Le, 2017; Zoph et al., 2017): train each architecture to converge to obtain separate $\boldsymbol{w}^*(\mathcal{P})$; (ii). Weight-sharing (Liu et al., 2018; Xie et al., 2018; Yao et al., 2020): share the same parameter $\boldsymbol{w}$ across different architectures during searching. And for both stand-alone and weight-sharing experiments, we repeat our method for five times and report the mean±std of test accuracy of searched architectures.

**Stand-alone setting.** To enable comparison under stand-alone setting, we use the NAS-Bench-201 dataset (Dong & Yang, 2020) where the authors exhaustively trained all subgraphs in a supernet and obtain a complete record of each subgraph's accuracy on three different datasets: CIFAR-10, CIFAR-100 and ImageNet-16-120 (details in Appendix E.1). We use the stochastic formulation and compare our method with (i) Random Search (Yu et al., 2020); (ii) REINFORCE (policy gradient) (Zoph & Le, 2017); (iii) BOHB (Falkner et al., 2018) and (iv) REA (regularized evolution) (Real et al., 2018). Results are in Table 2, and we can see that our method achieves better results than all existing stand-alone NAS methods and even finds the optimal architecture on CIFAR-10 and CIFAR-100 dataset.

**Weight-sharing setting.** We use the deterministic formulation, construct supernet follows (Liu et al., 2018), and evaluate all methods on CIFAR-10 dataset (details are in Appendix E.1). These are the

Table 3: Comparison with NAS methods in weight-sharing setting on CIFAR-10.

| Architecture | Test Error (%) | Params (M) | Search Cost (GPU Days) | Search Method |
|---|---|---|---|---|
| DenseNet-BC (Huang et al., 2017) | 3.46 | 25.6 | - | manual |
| ENAS (Pham et al., 2018) | 2.89 | 4.6 | 0.5 | RL |
| Random (Yu et al., 2020) | 2.85±0.08 | 4.3 | - | random |
| DARTS (1st) (Liu et al., 2018) | 3.00±0.14 | 3.3 | 1.5 | gradient |
| DARTS (2nd) | 2.76±0.09 | 3.3 | 4 | gradient |
| SNAS (Xie et al., 2018) | 2.85±0.02 | 2.8 | 1.5 | gradient |
| GDAS (Dong & Yang, 2019) | 2.93 | 3.4 | 0.21 | gradient |
| BayesNAS (Zhou et al., 2019) | 2.81±0.04 | 3.4 | 0.2 | gradient |
| ASNG-NAS (Akimoto et al., 2019) | 2.83±0.14 | 2.9 | 0.11 | natural gradient |
| NASP (Yao et al., 2020) | 2.83±0.09 | 3.3 | **0.1** | proximal algorithm |
| R-DARTS (Zela et al., 2020) | 2.95±0.21 | - | 1.6 | gradient |
| PDARTS (Chen et al., 2019) | 2.50 | 3.4 | 0.3 | gradient |
| PC-DARTS (Xu et al., 2020) | 2.57±0.07 | 3.6 | **0.1** | gradient |
| TRACE | 2.78±0.12 | 3.3 | 0.6 | gradient |
| TRACE + PC | **2.48±0.07** | 3.6 | 0.8 | gradient |

Table 4: Comparison with NAS methods in weight-sharing setting on ImageNet.

| Architecture | Top-1 Error (%) | Top-5 Error (%) | Params (M) | FLOPS (M) |
|---|---|---|---|---|
| ShuffleNet (Ma et al., 2018) | 25.1 | - | 5 | 591 |
| DARTS (Liu et al., 2018) | 26.7 | 8.7 | 4.7 | **514** |
| SNAS (Xie et al., 2018) | 27.3 | 9.2 | 4.3 | 522 |
| GDAS (Dong & Yang, 2019) | 26.0 | 8.5 | 5.3 | 581 |
| BayesNAS (Zhou et al., 2019) | 26.5 | 8.9 | **3.9** | - |
| PDARTS (Chen et al., 2019) | 24.4 | **7.4** | 4.9 | 557 |
| PC-DARTS (Xu et al., 2020) | 25.1 | 7.8 | 5.3 | 586 |
| TRACE | 26.5 | 8.6 | 4.7 | 520 |
| TRACE + PC | **24.2** | 7.7 | 5.0 | 563 |

most popular setups for weight-sharing NAS. Results are in Table 3 and 4, we can see that TRACE achieves comparable performances with existing weight-sharing NAS methods.

### 4.1.2 LOGIC CHAIN INFERENCE FROM KNOWLEDGE GRAPH (KG)

For logical chain inference, we use the deterministic formulation and compare our method with the following methods: Neural LP (Yang et al., 2017), DRUM (Sadeghian et al., 2019), and GraIL (Teru et al., 2020). Neural LP and DRUM are restricted to logical chain inference, while GraIL considers more complex graph structures. We also compare our method with random generated rules to better demonstrate the effectiveness of the proposed method. We do not compare our method with embedding-based methods, e.g. RotatE (Sun et al., 2019) as those methods all need embeddings for entities and cannot generalize found rules to unseen entities.

Following the setting of DRUM, we conduct experiments on three KG datasets: Family, UMLS and Kinship (details are in Appendix E.2), and report the best mean reciprocal rank (MRR), Hits at $1, 3, 10$ across 5 different runs. Results are in Table 5, which demonstrates that our proposed method achieves better results than all existing methods. Besides, case studies in Section 4.2 further demonstrate that TRACE can find more accurate rules than others.

### 4.1.3 META-PATH DISCOVERY IN HETEROGENEOUS INFORMATION NETWORK (HIN)

Finally, we apply TRACE to meta-path discovery problem on HINs. Following existing works (Wang et al., 2019; Yun et al., 2019), we use the deterministic formulation and conduct experiments on three benchmark datasets: DBLP, ACM and IMDB (details are in Appendix E.3) and compare our methods with the 1) baselines in GTN (Yun et al., 2019), i.e., DeepWalk (Bryan et al., 2014),

Table 5: Experiment results on logical chain inference.

| Dataset | Family | | | | UMLS | | | | Kinship | | | |
|---|---|---|---|---|---|---|---|---|---|---|---|---|
| | | Hits @ | | | | Hits @ | | | | Hits @ | | |
| | MRR | 10 | 3 | 1 | MRR | 10 | 3 | 1 | MRR | 10 | 3 | 1 |
| Neural LP | 0.88 | 0.98 | 0.95 | 0.80 | 0.72 | 0.93 | 0.84 | 0.58 | 0.55 | 0.87 | 0.63 | 0.41 |
| DRUM | **0.95** | **0.99** | **0.98** | 0.91 | 0.80 | **0.98** | 0.94 | 0.66 | 0.59 | **0.91** | 0.68 | 0.44 |
| GraIL | 0.87 | 0.96 | 0.95 | 0.79 | 0.73 | 0.94 | 0.86 | 0.55 | 0.57 | 0.89 | 0.65 | **0.45** |
| Random | 0.45 | 0.48 | 0.43 | 0.39 | 0.33 | 0.35 | 0.31 | 0.25 | 0.22 | 0.25 | 0.19 | 0.14 |
| TRACE | 0.92 | **0.99** | **0.98** | **0.92** | **0.81** | **0.98** | **0.95** | **0.67** | **0.61** | **0.91** | **0.69** | **0.45** |

Table 6: Evaluation results on the node classification task (F1 score).

| Dataset | DeepWalk | metapath2vec | GCN | GAT | HAN | GTN | Random | TRACE |
|---|---|---|---|---|---|---|---|---|
| DBLP | 63.18 | 85.53 | 87.30 | 93.71 | 92.83 | 93.98 | 91.62 | **94.42** |
| ACM | 67.42 | 87.61 | 91.60 | 92.33 | 90.96 | 91.89 | 90.28 | **92.62** |
| IMDB | 32.08 | 35.21 | 56.89 | 58.14 | 56.77 | 58.92 | 57.12 | **61.31** |

metapath2vec (Dong et al., 2017), GCN (Kipf & Welling, 2016), GAT (Veličković et al., 2018), HAN (Wang et al., 2019) and 2) random generated meta-paths. Results on different datasets are in Table 6, which demonstrate that TRACE performs better than other methods on different HINs.

## 4.2 CASE STUDY

To further investigate the performance of TRACE, we list the top rules found by TRACE and other methods in Table 7. This result demonstrates that our method can find more accurate logic rules than other baselines, which contributes to the superior performance of our method. We also give the architectures and meta-paths found by TRACE in Appendix F.

Table 7: An example of top 3 rules obtained by each method on Family dataset.

| Neural-LP | son(C, A) ← brother(C, B) , son(B, A) |
|---|---|
| | *son(B, A) ← brother(B, A)* |
| | son(C, A) ← son(C, B), mother(B, A) |
| DRUM | *son(C, A) ← nephew(C, B), brother(B, A)* |
| | son(C, A) ← brother(C, B), son(B, A) |
| | son(C, A) ← brother(C, B), daughter(B, A) |
| TRACE | son(C, A) ← son(C, B), wife(B, A) |
| | son(C, A) ← brother(C, B), son(B, A) |
| | son(C, A) ← nephew(C, B), brother(B, A) |

## 4.3 ABLATION STUDY

### 4.3.1 IMPACT OF ENCODING APPROACH

We compare TRACE with the following encoding methods on supernet: (i) DARTS, which introduces continuous parameters for each edge separately; (ii) RNN, which uses a RNN to compute the weights for each edge; (iii) CP decomposition, which generalizes DARTS to higher rank; (iv) TRACE(Full), which does not adopt Proposition 1 in Algorithm 1. Results on NAS-Bench-201 using CIFAR-100 are shown in Figure 3(a), and we can see that DARTS performs worse than other methods (CP and TRACE) due to insufficient expressive power. And TRACE achieves better results than CP by being topological aware. It also shows that our simplified encoding scheme does not harm the final performance, as verified in Proposition 1.

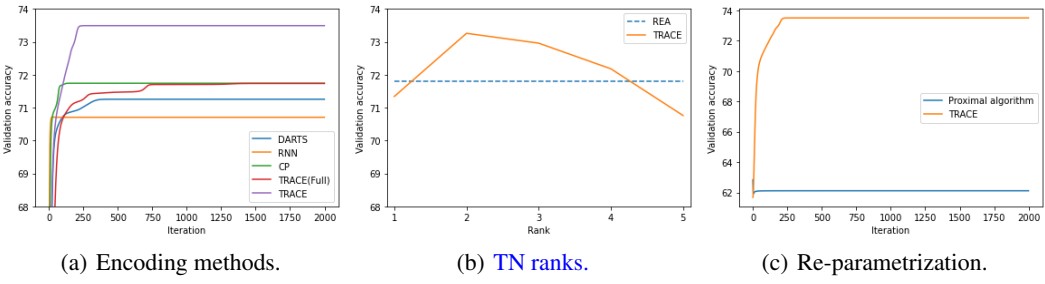

(a) Encoding methods.  (b) TN ranks.  (c) Re-parametrization.

Figure 3: Ablation studies on NAS-Bench-201, CIFAR-100 is used

### 4.3.2 IMPACT OF RANK OF TENSOR NETWORK

We also investigate the impact of $R_n$'s, which are ranks in tensor network (TN) on supernets. For simplicity, we restrict $R_n$ to be equal for all nodes $n \in \mathcal{N}(\mathcal{S})$ and compare the performance of different ranks with previous state-of-the-art REA (see Table 2) in Figure 3(b). Results demonstrate that while the rank can influence the final performance, it is easy to set rank properly for TRACE to beat other methods. We also adopt $R_n = 2$ for all other experiments.

### 4.3.3 OPTIMIZATION ALGORITHMS

Finally, we compare TRACE with proximal algorithm (Parikh & Boyd, 2014), which is a popular and general algorithm for constrained optimization. Specifically, proximal algorithm is used to solve (1) with the constraint $\sum_{i_-} \mathcal{T}_{i_-} = 1$ without Proposition 2. We solve the proximal step iteratively and numerically since there is no closed-form solutions. The comparison is in Figure 3(c), and we can see that TRACE beats proximal algorithm by a large margin, which demonstrates that the re-parameterized by Proposition 2 is useful for optimization.

## 5 CONCLUSION

In this paper, we *generalize* supernet from neural architecture search (NAS) to other machine learning tasks. To expressively model the supernet, we introduce a tensor formulation to the supernet and represent it by tensor network (*tensorizing supernets*). We further propose an efficient gradient-based algorithm to solve the new supernet search problem. Empirical results across various applications demonstrate that our approach has superior performance on these machine learning tasks.

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

## A COMPLETE ALGORITHMS OF OPTIMIZATION ON TENSORIZED SUPERNET

Here, we give a more detailed description for our algorithm under deterministic and stochastic formulation in Algorithm 3 and 4, respectively. In our experiments, we use the stochastic formulation for NAS under standalone setting, and deterministic formulation for others.

---

**Algorithm 3** TRACE (deterministic formulation with weight-sharing)

---

**Input:** Training set $\mathcal{D}_{\text{tr}}$, Validation set $\mathcal{D}_{\text{val}}$
1: Tensorize the supernet $\mathcal{T}$ with Algorithm 1;
2: Re-parameterize $\mathcal{T}$ to $\bar{\mathcal{T}}$ using (3);
3: **while** not converged **do**
4:     Update model parameters $\boldsymbol{w}(\beta)$ by gradient descending $\nabla_{\boldsymbol{w}}\mathcal{L}\left(f(\boldsymbol{w}, \beta), \mathcal{D}_{\text{tr}}\right)$
5:     Update supernet parameters $\beta$ by gradient ascending $\nabla_{\beta}\mathcal{J}_{\beta}$
6: **end while**
7: Obtain $\mathcal{P}^* = \mathcal{S}_{i_-}$ from the final $\bar{\mathcal{T}}$ by setting $i_- = \arg\max_{i_-} \bar{\mathcal{T}}_{i_-}$;
8: Obtain $\boldsymbol{w}^*(\mathcal{P}^*) = \arg\min_{\boldsymbol{w}}; \mathcal{L}\left(f(\boldsymbol{w}, \mathcal{P}^*), \mathcal{D}_{\text{tr}}\right)$ by retraining $f(\boldsymbol{w}, \mathcal{P}^*)$;
9: **return** $\boldsymbol{w}^*, \mathcal{P}^*$;

---

---

**Algorithm 4** TRACE (stochastic formulation)

---

**Input:** Training set $\mathcal{D}_{\text{tr}}$, Validation set $\mathcal{D}_{\text{val}}$
1: Tensorize the supernet $\mathcal{T}$ with Algorithm 1;
2: Re-parameterize $\mathcal{T}$ to $\bar{\mathcal{T}}$ using (3);
3: **while** not converged **do**
4:     Sample a subgraph $\mathcal{P}$ from the probability distribution given by $\bar{\mathcal{T}}$
5:     Solve $\boldsymbol{w}^*(\mathcal{P}) = \arg\min_{\boldsymbol{w}} \mathcal{L}\left(f(\boldsymbol{w}, \mathcal{P}), \mathcal{D}_{\text{tr}}\right)$
6:     Update supernet parameters $\beta$ by gradient ascending $\nabla_{\beta}\mathcal{J}_{\beta}$
7:     Save the best $\boldsymbol{w}^*(\mathcal{P}), \mathcal{P}$ so far
8: **end while**
9: **return** best $\boldsymbol{w}^*(\mathcal{P}), \mathcal{P}$ in Step 7;

---

## B COMPARISON OF PARAMETERS FOR DIFFERENT METHODS

Here, we compare the number of parameters and computational cost (FLOPs) of different methods for logic rule inference. We use $n$ as the length of logical chains, $d$ as the dimension of embeddings, $r$ as the rank of tensor networks and $e$ as the number of all relations. Results are in Table 8. Since $n$ and $r$ is often significantly smaller than $e$ and $d$ (typical values are $n = 3$, $r = 2$, $d = 128$ and $e \approx 20 - 30$), TRACE has comparable number of parameters and computational cost with existing methods.

Table 8: Comparison of number of parameters for different methods on KG.

| Method | # of parameters | FLOPs |
|---|---|---|
| Neural LP | $\mathcal{O}(d^2)$ | $\mathcal{O}(nd^2)$ |
| DRUM | $\mathcal{O}(rd^2)$ | $\mathcal{O}(nrd^2)$ |
| GraIL | $\mathcal{O}(ed^2)$ | $\mathcal{O}(edn)$ |
| TRACE | $\mathcal{O}(nr^2e)$ | $\mathcal{O}(nr^2e)$ |

For comparison on HIN, we use $n$ as the length of meta-paths, $d$ as the dimension of embeddings, and $r$ as the rank of tensor networks. Results are in Table 9. Note that $r$ is often relatively small compared with $d$ (typical values are $r = 2$ and $d = 128$), thus TRACE has the similar number of parameters and computational cost with existing methods.

Table 9: Comparison of number of parameters for different methods on HIN.

| Method | DeepWalk | metapath2vec | GCN/GAT/HAN | GTN | TRACE |
|---|---|---|---|---|---|
| # of parameters | $\mathcal{O}(nd)$ | $\mathcal{O}(nd)$ | $\mathcal{O}(nd^2)$ | $\mathcal{O}(n(r+d^2))$ | $\mathcal{O}(n(r^2+d^2))$ |
| FLOPs | $\mathcal{O}(n)$ | $\mathcal{O}(n)$ | $\mathcal{O}(n)$ | $\mathcal{O}(nr)$ | $\mathcal{O}(nr^2)$ |

Table 10: Comparison on number of parameters for different encoding methods.

| Method | Brief description |
|---|---|
| PDARTS | Progressively reduce number of operations and increase network depth |
| PC-DARTS | Use partial-connection in channels to reduce memory cost |
| DropNAS | Drop out some operations to avoid the Matthew effect |

## C    REVIEW ON EXISTING METHODS THAT CAN BE COMBINED WITH TRACE

## D    PROOFS

### D.1    PROPOSITION 1

*Proof.* Denote those nodes in supernet whose degree is greater than one (connected by more than one edges) as $\mathcal{N}'(\mathcal{S})$. Thus, $\mathcal{N}(\mathcal{S}) \setminus \mathcal{N}'(\mathcal{S})$ denotes all degree-1 nodes in supernet. Suppose one node $N_1(t)$ is only connected by edge $t$, while $N_2(t)$ is not, [3] then we can rewrite $\mathcal{T}_{i_-} = \sum_{r_n, n \in \mathcal{N}(\mathcal{S})}^{R_n} \prod_{t=1}^{T} \theta_{r_{N_1(t)}, i_t, r_{N_2(t)}}^{t}$ as follows:

$$\mathcal{T}_{i_-} = \sum_{r_n, n \in \mathcal{N}(\mathcal{S})} \prod_{t=1}^{T} \alpha_{r_{N_1(t)}, i_t, r_{N_2(t)}}^{t}$$

$$= \sum_{r_n, n \in \mathcal{N}'(\mathcal{S})} \sum_{r_n, n \in \mathcal{N}(\mathcal{S}) \setminus \mathcal{N}'(\mathcal{S})} \prod_{t=1}^{T} \alpha_{r_{N_1(t)}, i_t, r_{N_2(t)}}^{t}$$

$$= \sum_{r_n, n \in \mathcal{N}'(\mathcal{S})} \prod_{t=1}^{T} \sum_{r_n, n \in \mathcal{N}(\mathcal{S}) \setminus \mathcal{N}'(\mathcal{S})} \alpha_{r_{N_1(t)}, i_t, r_{N_2(t)}}^{t}$$

$$= \sum_{r_n, n \in \mathcal{N}'(\mathcal{S})} \dots \Big( \sum_{r_{N_1(t)}} \alpha_{r_{N_1(t)}, i_t, r_{N_2(t)}}^{t} \Big) \dots$$

$$= \sum_{r_n, n \in \mathcal{N}'(\mathcal{S})} \dots \tilde{\alpha}_{i_t, r_{N_2(t)}}^{(t)} \dots$$

where similar process is done for all edges. Thus, only nodes whose degree is greater than 1 is actually needed for index contraction. And for $n \in \mathcal{N}(\mathcal{S}) \setminus \mathcal{N}'(\mathcal{S})$, we can simply set $R_n = 1$ without loss of expressive power.    □

---

[3] If $N_1(t)$ is only connected by edge $t$ (degree is 1), $N_2(t)$ cannot be only connected by edge $t$ unless the supernet has only one edge $t$ connecting two nodes $N_1(t), N_2(t)$, which is a trivial case.

## D.2 PROPOSITION 2

*Proof.* Following (3), we can have:

$$
\sum_{i_-} \mathcal{T}_{i_-} = \frac{1}{\prod_{n \in \mathcal{N}'(\mathcal{S})} R_n} \sum_{i_-} \sum_{r_n, n \in \mathcal{N}'(\mathcal{S})}^{R_n} \prod_{t=1}^{T} \frac{\exp(\alpha_{r_{N_1(t)}, i_t, r_{N_2(t)}}^t)}{\sum_j \exp(\alpha_{r_{N_1(t)}, j, r_{N_2(t)}}^t)}
$$

$$
= \frac{1}{\prod_{n \in \mathcal{N}'(\mathcal{S})} R_n} \sum_{r_n, n \in \mathcal{N}'(\mathcal{S})}^{R_n} \prod_{t=1}^{T} \left( \sum_{i_t} \frac{\exp(\alpha_{r_{N_1(t)}, i_t, r_{N_2(t)}}^t)}{\sum_j \exp(\alpha_{r_{N_1(t)}, j, r_{N_2(t)}}^t)} \right)
$$

$$
= \frac{1}{\prod_{n \in \mathcal{N}'(\mathcal{S})} R_n} \sum_{r_n, n \in \mathcal{N}'(\mathcal{S})}^{R_n} 1^T
$$

$$
= \frac{1}{\prod_{n \in \mathcal{N}'(\mathcal{S})} R_n} \prod_{n \in \mathcal{N}'(\mathcal{S})} R_n = 1.
$$

$\square$

# E EXPERIMENT DETAILS

## E.1 NEURAL ARCHITECTURE SEARCH (NAS)

**Stand-alone setting** The supernet used in NAS-Bench-201 has 3 nodes, and each pair of nodes is connected by a directed edge, which gives 6 edges in total. And for each edge, we have 5 different operations ("choices"): zero, skip connect, $1 \times 1$ convolution, $3 \times 3$ convolution and $3 \times 3$ average pooling. And the details of datasets used in NAS-Bench-201 are in Table 11.

Table 11: Dataset statistics of three image classification dataset for NAS.

|  | Image size | # Classes | # Training | # Validation | # Test |
|---|---|---|---|---|---|
| CIFAR-10 | 32×32 | 10 | 25K | 25K | 5K |
| CIFAR-100 | 32×32 | 100 | 50K | 5K | 5K |
| ImageNet-16-120 | 16×16 | 120 | 151.7K | 3K | 3K |

**Weight-sharing setting** Our construction of supernet follows (Liu et al., 2018). The supernet has 7 nodes, where the first two nodes are the output from previous two cells, respectively, and the last node performs depthwise concatenation to the output of the rest four nodes. Thus, the supernet has 8 edges with multiple choices (operations), and for each edge, we consider 8 different operations: zero, skip connect, $3 \times 3$ and $5 \times 5$ separable convolution, $3 \times 3$ and $5 \times 5$ dilated separable convolution, $3 \times 3$ max pooling and $3 \times 3$ average pooling. We evaluate all weight-sharing methods on CIFAR-10 dataset and the dataset division is the same as in Table 11.

## E.2 LOGIC CHAIN INFERENCE

In our experiments, we follow the setting in DRUM (Sadeghian et al., 2019) and set the max length of rules $T$ to be 3 for all datasets. And we set the rank $L$ to be 4 in DRUM based on best validation performances. The details of KG datasets used in experiments are in Table 12.

Table 12: Dataset statistics for three KG dataset.

|  | # Triplets | # Relations | # Entities |
|---|---|---|---|
| Family | 28356 | 12 | 3007 |
| UMLS | 5960 | 46 | 135 |
| Kinship | 9587 | 25 | 104 |

## E.3 META-PATH DISCOVERY

The details of HIN datasets used in our experiments are in Table 13.

Table 13: Dataset statistics for three heterogeneous information network dataset.

|  | # Nodes | # Edges | # Edge types | # Features | # Training | # Validation | # Test |
|---|---|---|---|---|---|---|---|
| DBLP | 18405 | 67946 | 4 | 334 | 800 | 400 | 2857 |
| ACM | 8994 | 25922 | 4 | 1902 | 600 | 300 | 2125 |
| IMDB | 12772 | 37288 | 4 | 1256 | 300 | 300 | 2339 |

# F    MORE EXPERIMENT RESULTS

## F.1    CORRELATION ANALYSIS

Indeed, the correlation is a good criterion to show the rationality of one-shot architecture search methods (Bender et al., 2018; Liu et al., 2018; Yu et al., 2020; Guo et al., 2020). However, it is only a sufficient not necessary condition. Specifically, the goal of tensor $\mathcal{T}$ here is to capture good subgraph in the whole supernet, thus we expect the possibilities of $\mathcal{P}_i$ will concentrate on some top subgraphs, which is shown in below Figure 4.

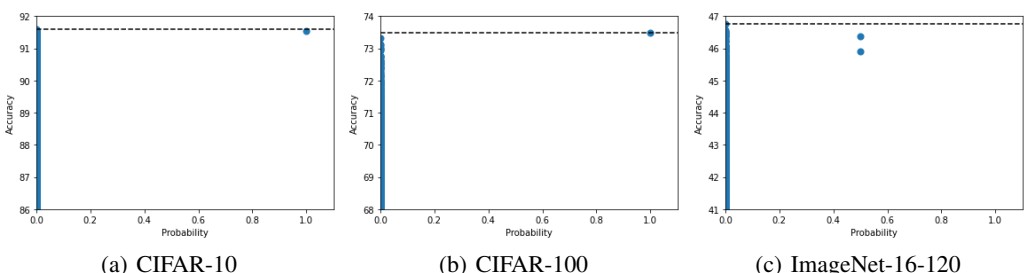

(a) CIFAR-10        (b) CIFAR-100        (c) ImageNet-16-120

Figure 4: Correlation of $\mathcal{T}$ and ground-truth accuracy on NAS-Bench-201 for different datasets.

## F.2    CASE STUDIES

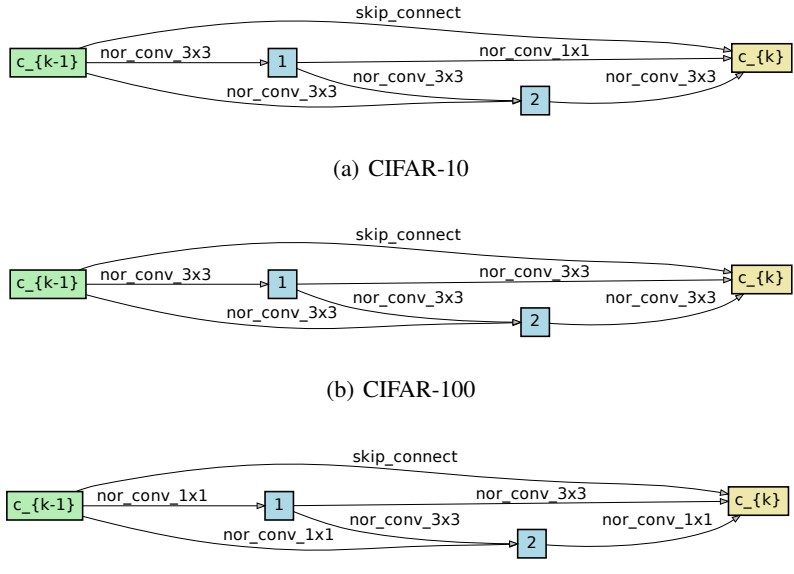

(a) CIFAR-10

(b) CIFAR-100

(c) ImageNet-16-120

Figure 5: Architectures found by TRACE on NAS-Bench-201 for different datasets.

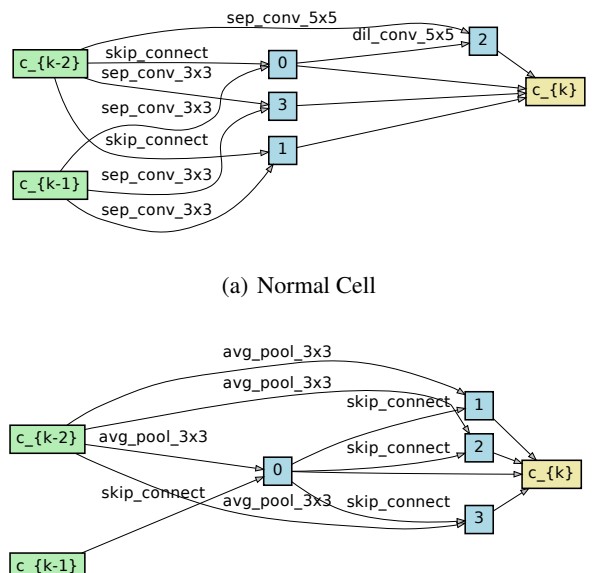

(a) Normal Cell

(b) Reduction Cell

Figure 6: Architectures found by TRACE on weight-sharing setting.

Table 14: Meta-paths found by GTN and TRACE on different HIN.

| Dataset | predefined meta-path | Top-3 meta-paths | |
| --- | --- | --- | --- |
| | | GTN | TRACE |
| DBLP | APCPA, APA | CPCPA, APCPA, CP | CPCA, APCPA, APA |
| ACM | PAP, PSP | APAP, APA, SPAP | PSP, APAP, PAP |
| IMDB | MAM, MDM | DM, AM, MDM | MDM, MDMAM, MAM |

## F.3 RUNNING PLOTS

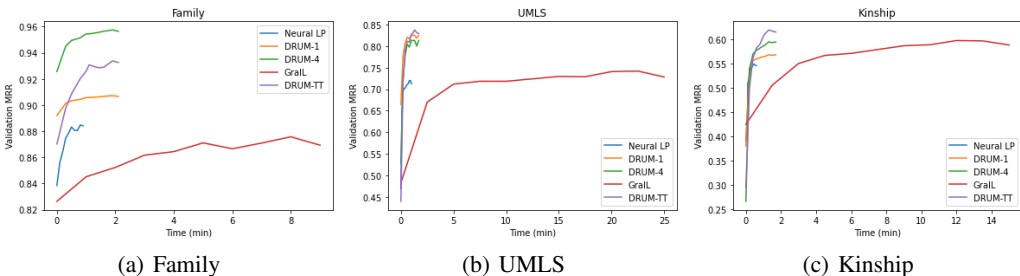

(a) Family        (b) UMLS        (c) Kinship

Figure 7: Validation MRR during training on three KG datasets.

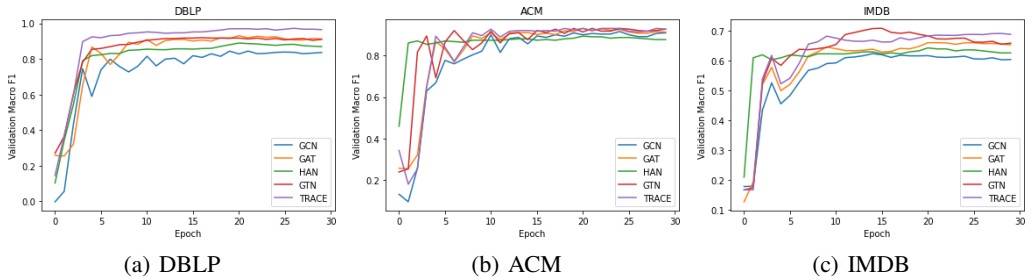

Figure 8: Validation macro F1 score during training on three HIN datasets.

## F.4 ABLATION STUDIES

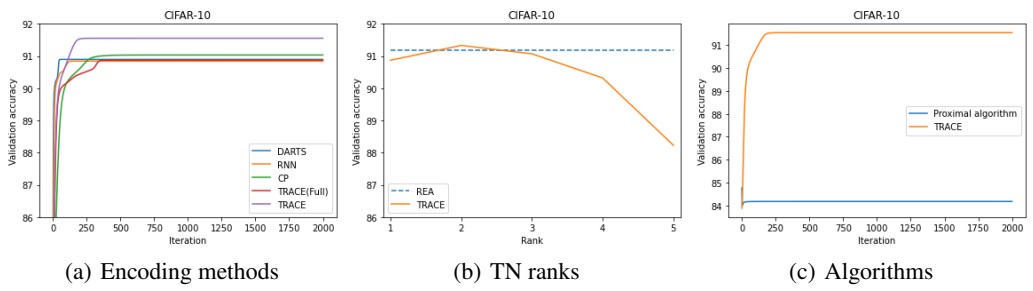

Figure 9: Ablation studies on NAS-Bench-201, CIFAR-10 is used

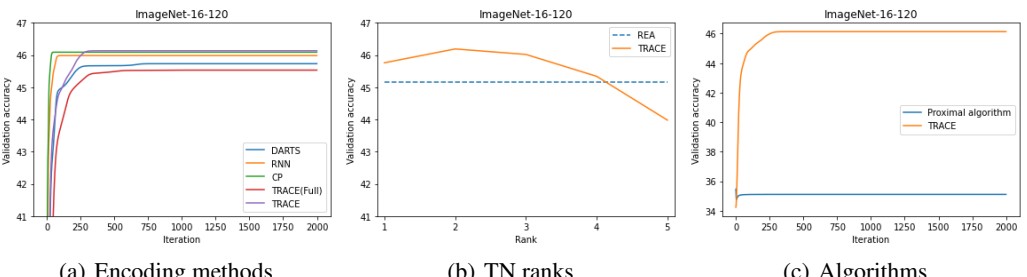

Figure 10: Ablation studies on NAS-Bench-201, ImageNet-16-120 is used

# G    ILLUSTRATION OF TENSORIZATION PROCESS FOR SUPERNETS

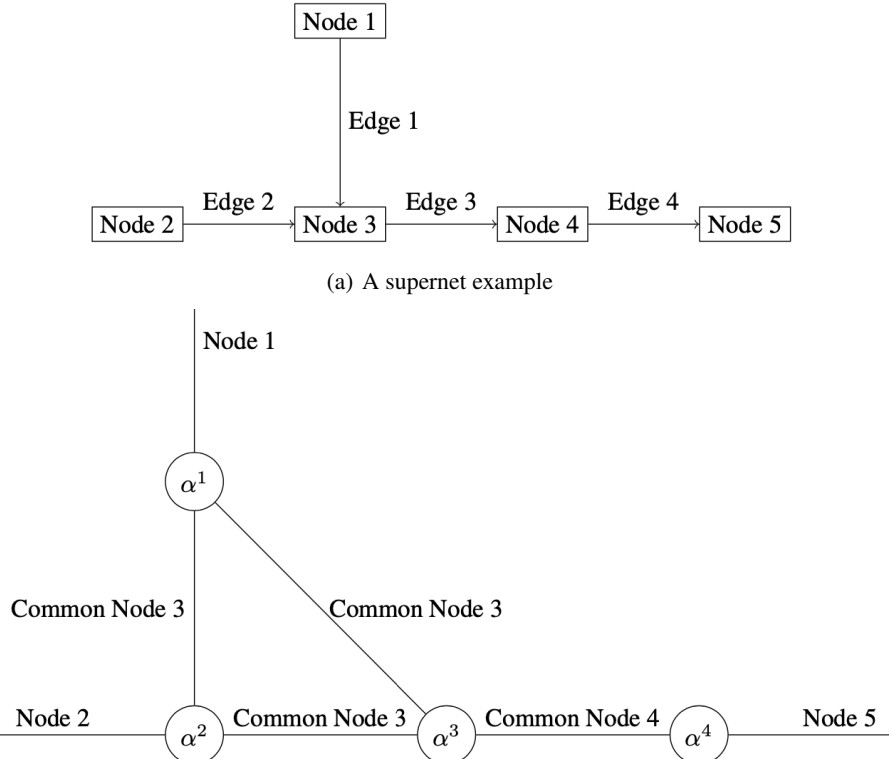

(a) A supernet example

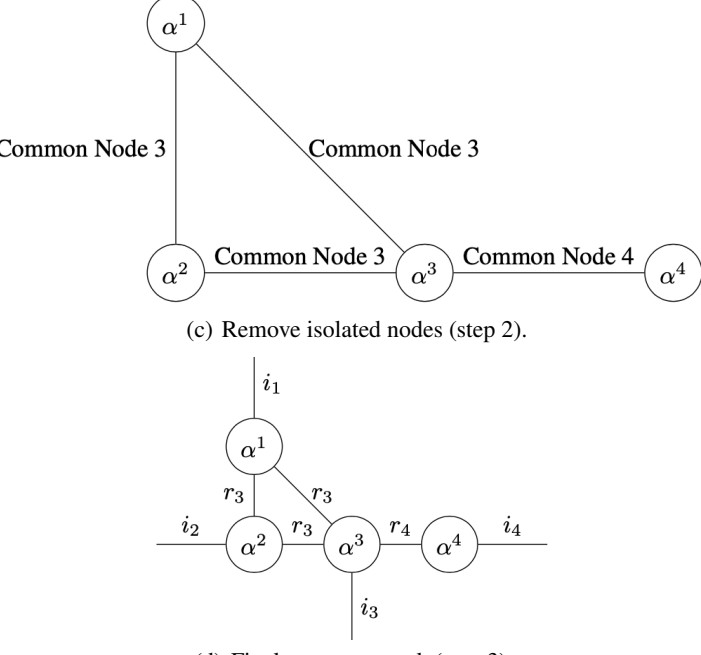

(b) Introduce $\alpha^t$ for each edge $t$, and connect them with nodes in the supernet (step 1).

(c) Remove isolated nodes (step 2).

(d) Final tensor network (step 3).

Figure 11: A step-by-step illustration of supernet encoding process (Algorithm 1).

