# OpenReview forum: "Generalizing and Tensorizing Subgraph Search in the Supernet"
_ICLR.cc/2021/Conference — Reject_

### Official Review · AnonReviewer2 · 2020-10-19

**Rating:** 5
**Confidence:** 3

**Review:**

This paper tries to use tensorize the supernet. The supernet itself is a graph and NAS aims to search the operations on the edges. The 'TRACE' finds that path-discovery appears in a number of tasks and NAS is one of them.

This is an interesting topic and the experiments are conducted on vision/logic chain/node classification tasks.

+) This paper is clear and easy to follow.

+) The topic is interesting. It is charming to design a universal method that could solve a number of problems.

However, there are still some concerns:

-) Is the title appropriate? The gradient-based NAS finds an optimal path in the global search space. However, for other tasks mentioned in this paper, is 'architecture search' appropriate to describe them? Could it be named as 'path search' or some others? Are there any other works support the 'architecture search' in other tasks?

-) The notations are not clear. The $\mathcal{J}$, the $r_{N_1(t)}$, the $\theta$ should all be described. However, I could not find them either in the 'annotation' paragraph or the first-mentioned sentences.

-) Move notations to the beginning of section 3.

-) What is $R_{N_1(t)}$ in Alg. 1? The $R$ has not been explained. However, I think this is important for the reader to understand the algorithm.

-) The results on CIFAR-10 aim to show that 'TRACE' could accelerate searching without loss of accuracy? What will happen if searching for longer epochs? According to 'Motivated by the success of previous methods (e.g., DARTS, SNAS and ENAS), our TN should include their encoding methods as special cases.', what is the number of parameters used for modeling the architectures?

-) ' their decomposition method does not consider the topological structure of supernets,'. For the cell-based NAS in Table. 1. Why the topological structure has not been considered?

-) In Fig. 2, it's better to add a toy example based on Eq. 4 and add more explanations.

-) Please report the cost in Table 4, 5. For example, the params/search cost/flops in Table 3.

-) (minor) Lack of experiments on large-scale datasets, like ImageNet.

-) The HIN task searches for the rank-1 CP (Table 1), and the CV task also searches for the rank-1 solution. So this HIN task is just adopting the standard solution on another dataset? Are there any other differences?

-) What if the rank goes higher in Figure 3 (b)?

Based on the quality of this paper, I don't think this paper creates something new. In my opinion, this paper describes the solutions from a novel/universal perspective. Some of the existing methods are special cases. So I selected 5 as the initial score. It would be nice if the authors emphasize the unique contribution more clear and address the concerns.

---

> ### Author Response · Authors · 2020-11-24
> **Response to Reviewer 2**
>
> Please first check the reply on the presentation and novelty in "common questions".
>
> Q1. The gradient-based NAS finds an optimal path in the global search space. However, for other tasks mentioned in this paper, is 'architecture search' appropriate to describe them? Could it be named as 'path search' or some others?
>
> We have revised the title and placed our submission in a better position. Please also refer to common question 1. Thanks for your great suggestion!
>
> Q2. The notations are not clear.
> - The $\mathcal{J}$, the $r_{N_1(t)}$, the $\theta$ should all be described. However, I could not find them either in the 'annotation' paragraph or the first-mentioned sentences.
> - Move notations to the beginning of Section 3.
> - What is $R_{N_1(t)}$ in Alg. 1? The $R$ has not been explained. However, I think this is important for the reader to understand the algorithm.
> - In Fig. 2, it's better to add a toy example based on Eq. 4 and add more explanations.
>
> Thank you for your suggestions. We have made thorough revisions (marked with blue) to our paper.
> - We have revised the notation part and move it to the beginning of Section 3
> - $R_{N_1(t)}$ and $R_{N_2(t)}$ are two hyper-parameters corresponding to the rank of tensor network.
> - Fig. 2 in section 3.2 is used to illustrate our encoding method, while Eq. 4 shows how to use softmax trick to satisfy the normalization constraint.
> Please refer to our main text for more detail.
>
> Q3. The results on CIFAR-10 aim to show that 'TRACE' could accelerate searching without loss of accuracy?
>
> Please have a double-check over Table 3. Our results do not support that TRACE can accelerate searching,
> it is NASP, PC-DARTS, and ASNG-NAS that support such a claim.
>
> Q4. What will happen if searching for longer epochs?
>
> Thanks for such an insightful question. Since what exactly will happen if searching a long time is still an open issue [A, B, C], we consider both stand-alone and one-shot setting. For a stand-alone setting, please see Figure 3(a),  which demonstrates that searching for longer epochs will have limited improvements; for the weight-sharing setting, our experimental setting follows the common setting in NAS literature, i.e. same training epochs.
>
> [A] Understanding and Robustifying Differentiable Architecture Search. ICLR 2020
>
> [B] Weight-Sharing Neural Architecture Search: A Battle to Shrink the Optimization Gap. arXiv preprint 2008.01475
>
> [C] Bridging the Gap between Sample-based and One-shot Neural Architecture Search with BONAS. NIPS 2020
>
> Q5. According to 'Motivated by the success of previous methods (e.g., DARTS, SNAS, and ENAS), our TN should include their encoding methods as special cases.', what is the number of parameters used for modeling the architectures?
>
> Thanks for your question. We have added a table (Table 8 in Appendix) to compare different modeling methods. Our TN uses a similar number of parameters with better performances.
>
> Q6. 'their decomposition method does not consider the topological structure of supernets,'. For the cell-based NAS in Table. 1. Why the topological structure has not been considered?
>
> Previous results apply the same encoding method (CP decomposition) for different supernets, which does not consider their topological structures. We have revised our paper to make our logic more clear. Please refer to the revision (marked with blue) in Section 3.
>
> Q7. Please report the cost in Table 4, 5. For example, the params/search cost/flops in Table 3.
>
> Thanks for your suggestion. Table 4 and 5 follow existing works [Sadeghian et al., 2019, Yun et al., 2019] and it is not a common practice in KG/HIN area to report these numbers. The reason for that is these numbers are quite similar for all compared methods. And we also add tables (Table 8 and 9 in Appendix B) to show the similar computational cost of compared methods.
>
> Q8. Lack of experiments on large-scale datasets, like ImageNet.
>
> Thanks for your suggestion. We have added the experiments on ImageNet in Table 4. Our experimental results demonstrate that TRACE+PC can achieve better performances than previous NAS methods.
>
> Q9. The HIN task searches for the rank-1 CP (Table 1), and the CV task also searches for the rank-1 solution. So this HIN task is just adopting the standard solution on another dataset?
>
> Thanks for admitting our conceptual novelty. We conceptually unify these applications and show they can be represented by a tensor formulation. And we construct a tensor network to replace the CP decomposition, which can exploit the topological structure of supernets for better performances. This novel discovery by us leads to such a misunderstanding.
>
> Q10. What if the rank goes higher in Figure 3 (b)?
>
> Thanks for your suggestion. We have made the rank go higher in experiments and the results are added to Figure 3(b). From the figure, we can see that when the rank goes higher, the performance becomes worse due to over-parametrization and difficult optimization.

---

### Official Review · AnonReviewer3 · 2020-10-27
**A method that generalizes the idea from NAS to general tasks**

**Rating:** 4
**Confidence:** 5

**Review:**


+ This paper generalizes the supernet search problem on a broader horizon. Specifically, some of the current NAS methods use supernet to co-training different neural architectures for further architecture search. This paper does not just consider supernet as a tool for NAS, but also consider supernet as a graphical model and extend supernet to several general tasks in the form of graph data. (+)

+ This paper unifies the above tasks by a tensor formulation and encodes the topology inside the supernet by a tensor network. Different from NAS, the topological structure of the supernet is utilized in this paper. (+)

+ The paper further proposes an efficient algorithm that admits both stochastic and deterministic objectives to solve the search problem. (+)

+ A wide range of machine learning tasks besides computer vision are used to evaluate the proposed method's effectiveness, such as logic chain inference on knowledge graphs and meta-path discovery for HIN. (+)

- After reading the related work section, we notice that the authors have known some methods used in the NAS problem, but they are not aware of the current NAS's ineffectiveness problem. Specifically, the NAS's effectiveness is open to question, with its architecture rating is often inaccurate, especially in DARTS and some single-path one-shot NAS method. Two ICLR 2020 papers suggest that many NAS methods are not better than random architecture search. This paper borrows the concept from NAS but ignore the ineffectiveness/inefficiency of NAS. Therefore, the proposed method's effectiveness in this paper is questionable unless the authors provide an architecture rating analysis. (-)

In the context of chaotic phenomena in NAS nowadays, analyzing the architecture rating problem is of most importance. As there are many NAS papers published every year and their ineffectiveness may still be not widely recognized by the reviewers and the public, merely borrowing NAS methods to other domains (e.g., in this paper) is dangerous. This may cause more chaos in the new field. I think a NAS method's effectiveness should be sufficiently investigated before NAS can be generalized to a new domain.


- As defined, Ti in [0, 1] represents how ""good"" P can be. Then, there must be a connection between T and M(f(w, P), D), which represents the performance of learning model f(w, P) on dataset D? My understanding is that the authors can use the CORRELATION BETWEEN EDGE IMPORTANCE AND MODEL ACCURACY in the following paper [reference] to measure the connection between T and M. In other words, as is formulated by the authors, "Ti ∈ [0, 1] represent how "good" P can be," can the authors provide pieces of evidence showing the proposed method can actually have this effect? This is exactly related to the inaccurate architecture rating problem in NAS. Only can the authors answer this question will I be able to consider an acceptance of this paper. (-)


[reference]
@inproceedings{
anonymous2021dots,
title={{\{}DOTS{\}}: Decoupling Operation and Topology in Differentiable Architecture Search},
author={Anonymous},
booktitle={Submitted to International Conference on Learning Representations},
year={2021},
url={https://openreview.net/forum?id=y6IlNbrKcwG},
note={under review}
}


+ The way to use routine (Algorithm 1) to tensorize the supernet is reasonable and correct. （+）


- This paper is very difficult to understand. I must read very carefully (especially for the subscript notation) twice until I can understand the paper. I think the main idea of this paper is simple. But the authors use a complicated way to describe the method, which makes the article unreadable and thus mysterious. (-)

Overall, this paper proposes a method that generalizes the idea from NAS to general tasks. However, it is open to question whether this generalization is appropriate as the supernet-based NAS's effectiveness is an active research topic.

---

> ### Author Response · Authors · 2020-11-25
> **Reply to Reviewer 3**
>
> Please first check the reply on the presentation and novelty in "common questions".
>
> Q1. The authors are not aware of the current NAS's ineffectiveness problem. Specifically, the NAS's effectiveness is open to question, with its architecture rating is often inaccurate, especially in DARTS and some single-path one-shot NAS method. Two ICLR 2020 papers and many ICLR 2021 submissions suggest that many NAS methods are not better than random architecture search. This paper borrows the concept from NAS but ignore the ineffectiveness/inefficiency of NAS. Therefore, the proposed method's effectiveness in this paper is questionable...
>
> Thanks for mentioning such concerns,
> the validity of TRACE is as follows:
> - First,
> please check common question 1,
> this submission is an algorithmic paper that targets at the supernet search problem instead of NAS.
> - Second,
> we are indeed deeply aware of the problems you mentioned in NAS.
> Thus,
> we perform experiments on both the stand-alone setting
> and one-shot setting.
> Your concerns
> fall into the one-shot setting,
> and are eliminated in the stand-alone one.
> However,
> the effectiveness of TRACE is supported by both settings.
> - Third,
> the superiority of TRACE
> is further cross-validated by other applications from HIN and KG
> compared with random search.
>
> Q2. ... unless the authors provide an architecture rating analysis. As defined, $\mathcal{T}$ represents how "good" $\mathcal{P}$ can be. Then, there must be a connection between $\mathcal{T}$ and $\mathcal{M}(f(w, \mathcal{P}), \mathcal{D})$, which represents the performance of learning model $f(w, \mathcal{P})$ on dataset $\mathcal{D}$? My understanding is that the authors can use the CORRELATION BETWEEN EDGE IMPORTANCE AND MODEL ACCURACY in the following paper [reference] to measure the connection between $\mathcal{T}$ and $\mathcal{M}$. In other words, as is formulated by the authors, ``$\mathcal{T}_{i-} \in (0,1)$ represent how "good" P can be, can the authors provide pieces of evidence showing the proposed method can actually have this effect? This is exactly related to the inaccurate architecture rating problem in NAS.
>
> Thanks for your suggestion.
> Indeed,
> correlation is a good criterion to show the rationality of
> one-shot architecture search methods [Bender et al., 2018; Liu et al., 2018; Yu et al., 2020; Guo et al., 2020].
> However,
> it is only a sufficient, not necessary condition.
> Specifically,
> the goal of tensor $\mathcal{T}$ here is to capture good
> subgraph in the whole supernet,
> thus we expect the possibilities of $\mathcal{P}_i$ will concentrate on some top
> subgraphs,
> which is shown in below Figure 4 (Appendix F.1).
> Thus,
> TRACE can accurately capture top subgraphs in the supernet.
>
> Q3. This paper is very difficult to understand...
>
> Thanks for your suggestion.
> We have rewritten our methodology part to make it more clear.
> Please refer to the blue-marked text in our paper for revision.

---

### Official Review · AnonReviewer4 · 2020-10-28
**Official Blind Review #4**

**Rating:** 5
**Confidence:** 3

**Review:**

Summary:

This paper proposes a new and general formulation for supernet, which encodes supernet with tensor network(TN). Based on TN, the topology of supernet can be encoded. Besides, this paper proposes a corresponding algorithm to solve the search problem.

Reasons for score:

Overall, I vote for accepting marginally. My major concern is about the clarity of the paper and some additional experiments (see cons below). Hopefully the authors can address my concern in the rebuttal period.

Pros:

1.In this paper, the authors introduce a new and general formulation for supernet. Due to its generalization, supernet-based NAS methods with it can be applied to many deep learning tasks. In addition, this formulation can encode the topology of supernet, which benefits the network search.

2.Based on this formulation, this paper proposes a corresponding search algorithm. This algorithm can solve supernet search problem for both deterministic formulation and stochastic formulation.

3.Experiments are well thought out and highlight the key advantages of the method over other NAS methods.

Cons:
1.For recent NAS methods, they all validate their performances on ImageNet. In the experiments, there is only a experiment on ImageNet-16-120. Could you provide an experiment in weight-sharing setting on ImageNet dataset?

2.For Table 3, it is a must to compare with more NAS methods, such as PDARTS and DropNAS. In this way, the effective of the proposed method can be validated fully.

3.In section 3.2, $R^{n}$ in $\mathcal{T}_{i-}$ is used without any definition. Are there some hints for it? If not, it would lead to some misunderstandings.

Questions during rebuttal period:

Please address and clarify the cons above

Some typos:

Introduction, Notations, last line: a vector $\textbf{o}_{i} \in \mathbb{R}^{n} $ -> $\textbf{o} \in \mathbb{R}^{n} $

---

> ### Author Response · Authors · 2020-11-24
> **Reply to Reviewer 4**
>
> Please first check the reply on the presentation and novelty in "common questions".
>
> Q1. For recent NAS methods, they all validate their performances on ImageNet. In the experiments, there is only an experiment on ImageNet-16-120. Could you provide an experiment in weight-sharing setting on ImageNet dataset?
>
> Thank you for your suggestion.
> We have added the experiments on ImageNet in Table 4.
> Results demonstrate that TRACE+PC can achieve better
> performances than previous NAS methods.
>
> Q2. For Table 3, it is a must to compare with more NAS methods, such as PDARTS and DropNAS. In this way, the effectiveness of the proposed method can be validated fully.
>
> Thank you for your suggestion.
> Please first see the position of our paper in common question 1.
> The word "compare" here is not that accurate
> as PDARTS and DropNAS solve the NAS problem
> in an orthogonal way to ours.
> However,
> we also add the performance of recent NAS methods and TRACE combined with
> PC-DARTS in Table 3,
> which demonstrates that our method can be combined with recent NAS
> to achieve better performances.
>
> Q3. In Section 3.2, $R_n$ in $\mathcal{T}_{i-}$ is used without any definition. Are there some hints for it? If not, it would lead to some misunderstandings.
>
> The $R_n$ here is the rank of our proposed tensor network.
> To avoid ambiguity, we have also rewritten the methodology part.
> Please see our response to common question 2
> and our revision (marked with blue) in our paper.

---

### Official Review · AnonReviewer1 · 2020-10-31
**This paper proposed a new way to unify the searching strategy for NAS**

**Rating:** 5
**Confidence:** 3

**Review:**

The idea of employing a tensor method to generalize NAS seems to be interesting. However, this paper suffers from a few problems:
1. The presentation of this paper requires improvement. There are some confusion:

- This paper has not clearly described the difference with previous works. Specifically, although this paper claims to give a unified method, it is difficult to distinguish what have been unified. And in the 10-th line of the Abstract, what is the concrete description of "the above problems"?
- In Figure 2,  whether $\alpha^{(1)}$ is the $\theta^{1}$? If yes, in Figure 2(b), it would be better to use $\theta^1$ to replace $\alpha^{(1)}$ directly. And in the caption of Figure 2, what does "$N_1(t)/N_2(t)$" mean, a division or "$N_1(t)$ or $ N_2(t)$"?
- In Algorithm 1, what is the meaning of $R_{N_1(t)}$? And how to choose the value of $R_{N_1(t)}$?
- In Section **PROPOSED METHOD**, how to transform a supernet to a tensor graph?  It is highly suggested that adding more details about the transforming process.

2. The experimental results are not good enough. For example,  in results on Cifar10, there are a number of competitive methods which obtained better results of the proposed one.

| Model                                           | Err.      | #Params | Cost(GPU days) |
| ----------------------------------------------- | --------- | ------- | -------------- |
| TRACE(This paper)                               | 2.78±0.12 | 3.3     | 0.6            |
| P-DARTS + cutout(Xin Chen et. al.. ICCV 2019.)  | 2.5       | 3.4     | 0.3            |
| PC-DARTS + cutout(Yuhui Xu et. al.. ICLR 2020.) | 2.57±0.07 | 3.6     | 0.1            |

3. The writing requires significant proofreading. There are plenty of grammar errors：

- In the 2-nd line of Section 2.2, "Tensor methods have found wide applications" -> "... have been  found in..."
- In the 5-th line of Section 3.1, "Note that a subnet P can be distinguished by its choices on each edge, we propose...." -> "... each edge, and we propose...."
- In page 2,  "it also emerge in"
- etc..

---

> ### Author Response · Authors · 2020-11-24
> **Reply to Reviewer 1**
>
> Please first check the reply on the presentation and novelty in "common questions".
>
> Q1. This paper has not clearly described the difference with previous works.
> 	Specifically, although this paper claims to give a unified method,
> 	it is difficult to distinguish what has been unified.
> 	And in the 10th line of the Abstract, what is the concrete description of "the above problems"?
>
> We have unified different objects (neural architecture, logic rule, meta-path)
> that originally arise in different areas as a ````"generalized architecture" in supernet.
> Based on this unified treatment, we propose a novel algorithm to solve
> this ``"generalized architecture search" problem.
> We have rewritten the abstract and methodology part to emphasize our contribution
> on a unified framework.
>
> Q2. In Figure 2, whether is $\alpha^{(1)}$ the $\theta^1$?
> 	If yes, in Figure 2(b),
> 	it would be better to use $\theta^1$ to $\alpha^{(1)}$ replace directly.
> 	And in the caption of Figure 2,
> 	what does "$N_1(t)/N_2(t)$" mean,
> 	a division or ``"$N_1(t)$ or $N_2(t)$"?
>
> We have changed ``$\theta$ to $\alpha$ and "$N_1(t)/N_2(t)$" to "`$N_1(t)$ or $N_2(t)$" to avoid ambiguity.
>
> Q3. What is the meaning of $R_{N_1(t)}$? And how to choose the value of $R_{N_1(t)}$?
>
> $R_{N_1(t)}$ is a hyper-parameter in our method which corresponds to the rank of the tensor network.
> We have thoroughly rewrite the methodology part (common question 2)
> and marked important revisions with blue color.
> And we have done an ablation study on the rank in Section 5.3.2,
> which demonstrates that the optimal rank should be set based on experiments.
>
> Q4. In Section PROPOSED METHOD, how to transform a supernet to a tensor graph? It is highly suggested that adding more details about the transforming process.
>
> Thank you for your suggestion.
> We have re-written Section 3.2.
> Specifically, we introduce a third-order tensor $\alpha^t$ for each edge,
> which is based on previous methods (e.g., DARTS and SNAS),
> but uses tensors instead of vectors to allow more flexibility.
> And we use index summation to reflect the topological structure (common nodes)
> between different edges.
> The transformation from this computation process to a ``tensor graph''
> follows the common practice in tensor decomposition literature,
> please refer to [A, B] for more detail.
>
> Besides, we have added more details to make our methods more clear (a graphical illustration is added to Appendix.G).
>
> [A] Tensor networks for dimensionality reduction and large-scale optimization: Part 1 low-rank tensor decompositions. Foundations and Trends in Machine Learning, 2016.
>
> [B] Tensor networks for dimensionality reduction and large-scale optimization: Part 2 applications and future perspectives. Foundations and Trends in Machine Learning, 2017.
>
> Q5. The experimental results are not good enough. There are a number of competitive methods that obtained better results than the proposed one on Cifar10.
>
> Thank you for your suggestion.
> We have added more results of recent NAS methods on CIFAR-10 in Table 3.
> Please also refer to common question 1,
> where we discuss the relation between TRACE and other NAS methods.
> And we can achieve better performances than recent NAS methods
> by combining TRACE with PC-DARTS, as is also shown in Table 3.
>
> Q6. The writing requires significant proofreading. There are plenty of grammar errors...
>
> We have thoroughly revised all possible errors. Thank you for your tolerance.

---

### Decision · Program_Chairs · 2021-01-07
**Final Decision**

**Decision:**

Reject

**Comment:**

This paper proposes a new and general formulation for supernet, which encodes supernet with tensor network(TN). The idea is interesting and motivated.  However, the paper is well presented and the clarify needs to be further improved.  The effectiveness of algorithm is not well justified and experimental results are less convincing even after additional results provided in the revision. Most importantly, it is not clear that the 'TENSORIZING' method can solve the current NAS's ineffectiveness problem.  It is confirmed that the reference to ICLR-2021 paper is not used for the decision of paper.